



**Rainfall redistribution in subtropical Chinese forests changes over 22**
**years**
Wanjun Zhang[a,b], Thomas Scholten[c], Steffen Seitz[c], Qianmei Zhang[a,b], Guowei Chu[a,b], Linhua
Wang[a,b], Juxiu Liu[a,b,*]
[a]*Key Laboratory of Vegetation Restoration and Management of Degraded Ecosystems, South China*
*Botanical Garden, Chinese Academy of Sciences, Guangzhou, 510650, China*
[b]*Guangdong Provincial Key Laboratory of Applied Botany, South China Botanical Garden, Chinese*
*Academy of Sciences, Guangzhou, 510650, China*
[c]*Soil Science and Geomorphology, Department of Geosciences, University of Tübingen, Tübingen,*
*72070, Germany*
[*] Correspondence to: Juxiu Liu (ljxiu@scbg.ac.cn)





**Abstract**
Rainfall redistribution through the vegetation canopy plays a key role in the
hydrological cycle. Although there have been studies on the heterogeneous patterns of
rainfall redistribution in some ecosystems, the understanding of this process in different
stages of forest succession remains insufficient. Therefore, this study investigated the
change tendency of rainfall redistribution and rainwater chemistry in a subtropical
succession forest area in South China, based on 22 years (2001–2022) of monitoring
740 valid rainfall events. Results showed that at the event scale throughfall ratio showed
in order broadleaf forest (BF) < mixed forest (MF) < pine forest (PF), and stemflow
ratio showed in order BF > MF > PF. At the interannual scale, annual gross rainfall
considerably shifted over time, which directly induced the variable accumulation of
annual throughfall and annual stemflow. In the last 22 years, annual variability of
throughfall presented in order MF (*CV*, 9.7%) < BF (15.6%) < PF (16.1%), and annual
variability of stemflow presented in order MF (*CV*, 38.6%) < PF (50.9) < BF (56.2%).
The spatial variability of stemflow was always greater than that of throughfall. Besides,
the difference of rainwater chemistry fluxes (TN, TP and $K^+$) among the three forest
types were found and they changed over time. Throughfall was characterized with high
chemistry fluxes compared to open rainfall followed by stemflow. On average, TN and
TP fluxes of throughfall presented in order BF < MF < PF, while $K^+$ flux of throughfall
presented in order BF > MF > PF. The above results indicated that the patterns of
rainfall redistribution often changed over time. The event-scale accumulation of
throughfall and stemflow potentially induced the interannual-scale variability. Both
water volume and chemistry of throughfall and stemflow depended on the effect of
rainfall and forest factors. This study provided insight into the rainfall redistribution
process by linking the long-term changing of rainfall pattern and subtropical forest
succession sequence.
**Keyword**: throughfall, stemflow, variability, forest types, long-term



## 1. Introduction

In recent years, there has been on-going concern about the potential impacts of climate change on forest ecosystems, particularly in terms of rainfall input associated to water resource (Reynaert et al., 2020; Grossiord et al., 2017; Bruijnzeel et al., 2011; Leuzinger and Körner, 2010). Numerous studies have documented rainfall regimes and their effect on the water cycle in different regions of the world, including spatial and temporal changes in the amount, intensity, and frequency (Brasil et al., 2018; Ponette-González et al., 2010). Meanwhile, these variables in rainfall refer to the redistribution of rainfall into canopy interception, throughfall and stemflow, being important components of ecosystems hydrological processes (Germer et al., 2010; Levia and Frost, 2006; Loustau et al., 1992). Rainfall redistribution patterns can impact the biogeochemistry cycle by affecting soil moisture, which in turn affects the activity of soil microorganisms that decompose organic matter (Tonello et al., 2021a; Junior et al., 2017; Van Stan II and Pypker, 2015). The study of Sun et al. (2023) verified that throughfall reduction significantly affected soil carbon cycle in a subtropical forest. Therefore, understanding the roles of rainfall redistribution in the water cycle is essential.

Rainfall redistribution, as the partitioning into interception loss, throughfall and stemflow, is an important hydrological process that regulates water and nutrient cycling in forest ecosystems. Interception loss refers to the part of the event rainfall intercepted by the canopy, accounting for about 10%–30% of gross rainfall depending on the studied forest canopy, such as shrub (Zhang et al., 2015), mixed broadleaf (Yan et al., 2003), pine (Loustau et al., 1992). This portion of the rainwater evaporates directly back into the atmosphere. Later on, the remaining rainwater reaches the ground either as throughfall or stemflow. Throughfall is a critical component of rainfall redistribution, and it on average contributes to approximately 60%–90% of the gross rainfall on the floor in forests, shrubland or cropland. (Zhang et al., 2023; Zhang et al., 2021; Brauman et al., 2010; Marin et al., 2000). Raindrops coalesce or splash on canopy leaf surfaces, generating spatially different throughfall volume and raindrop kinetic energies which can be larger or lower than that of open rainfall (Levia et al., 2019; Goebes et al., 2015). Stemflow, the left rainwater flowing bottomwards along the plant stem or trunk, often accounts for only a small proportion (0–12%) of rainfall (Niu et al., 2023; Yue et al, 2021; Llorens and Domingo 2007). Nevertheless, stemflow inputs can be important as hot spots for near-trunk soils, inducing water and nutrient enrichment and deep



infiltration, but also erosion (Zhao et al., 2023; Llorens et al., 2022). It can funnel more water than open rainfall on an equivalent area and contributes to 10% of the annual soil water input (Levia and Germer, 2015; Chang and Matzner, 2000). Throughfall and stemflow restrict water input to the soil layer, thereby affecting soil moisture conditions, runoff generation and water and nutrient cycling (Lacombe et al., 2018; Klos et al., 2014).

The proportions of rainfall redistribution are generally driven by meteorological conditions (e.g., rainfall amount, intensity, duration) and vegetation cover (e.g., canopy structures, tree characteristics) (Tonello et al., 2021a; Sun et al., 2018; Mużyło et al., 2012; Nanko et al., 2006). For meteorological conditions, numerous studies have documented that throughfall volume and stemflow volume increase with increasing gross rainfall and intensity (Ji et al., 2023; André et al., 2011). The ratios of throughfall and stemflow were both characterized with logarithmically increasing with rainfall, tending to be quasi-constant for heavy rainfall events (Zhang et al., 2021; Liu et al., 2019). This was also synchronously related to the gradual saturation of the canopy which limited the ratios of rainwater partitioning (Carlyle-Moses et al., 2004). Besides, differences of water volume spatially exist from place to place. The spatial variability (expressed as a coefficient of variation) of throughfall volume is generally higher for small rainfall events (< 10 mm) than that for heavy rainfall events (Germer et al., 2006; Price et al., 1997).

Rainfall redistribution among different plant species can vary significantly due to differences in the structure and characteristics of their canopies. In special, some of the key factors determine the redistribution of rainfall, for example, leaf area index (LAI), leaf shapes and orientations can affect the amount of intercepted loss and throughfall (Zhang et al., 2021; Goebes et al., 2015; Keim et al., 2006). The diameter at breast height (DBH), bark type and orientation of trunks/stems and branches influence the amount of stemflow (Levia et al., 2015; Livesley et al., 2014; Germer et al., 2010). For each of the rainfall partitioning fluxes, their responses to the influential predictors often show high variation. A modelling study of rainfall partitioning in China explained that throughfall was best represented by mean tilt angle (MTA), followed by DBH. Subsequently, DBH was the dominant predictor for stemflow, followed by LAI and bark texture (Zhang et al., 2023). Due to these factors, rainfall redistribution presented different degrees of spatial variability. This variability (expressed as coefficient of variation) decreased with increasing rainfall amount and intensity, consequently



tending to be quasi-constant (Germer et al., 2006). Besides, at interannual scale, ratios
of rainfall redistribution are driven by annual canopy structures. The study of Niu et al.
(2023) documented that annual throughfall ratio gradually increased, while annual
stemflow ratio and interception loss ratio decreased with increasing thinning intensity
in shrub plantation. Meanwhile, annual changes of rainfall events (amount and intensity)
reinforced the time instability of throughfall spatial variability (Rodrigues et al., 2022).
Overall, the rainfall-canopy interactions play a key role in rainfall redistribution
processes and further affect the water cycle in many ecosystems.

Vegetation canopy is the functional interface between ecosystem and atmospheric

wet deposition (Van Stan II and Pypker, 2015). The leaf and trunk/stem, acting as a
filter, alter rainwater chemical concentrations via leaching and depositing processes. As
a result, throughfall and stemflow exhibit high chemical concentrations compared to
the open rainfall (Jiang et al., 2021; Zimmermann et al., 2007). For instance, in a Chinese
pine plantation, the volume weighted mean concentrations of $NH4^+$ and $NO3^-$ in
throughfall were significantly higher than those in open rainfall (Wang et al., 2023).
Stemflow ion fluxes (e.g., $K^+$) from deciduous tree species were greater than those for
evergreen tree species because of the differences in bark morphology and branch
architecture (Su et al., 2019). Moreover, it is also common that throughfall and
stemflow chemistry fluctuated seasonality with the shifts in rainfall regime and leaf
growth (Turpault et al., 2021; Siegert and Levia, 2014; Staelens et al., 2007). A large
number of elements required by plants are mainly N, P, K, Ca and Mg. In general, N, K
and Ca are the most important inputs to forest ecosystem, and P is the least. Phosphorus
(P) is considered to be a limiting nutrient element in tropical and subtropical forests.
The long-term productivity of vegetation depends on the input of atmospheric P.
Besides, the increasing trend in seasonal drought and atmospheric nitrogen (N)
deposition in subtropical areas of China were reported (Zhou et al., 2011), and may
inhibit the growth of plant and affect productivity and functioning of forest ecosystems
(Wu et al., 2023; Borghetti et al., 2017). Therefore, the important significance of
atmospheric precipitation to the ecosystem can also be seen from the amount of element
cycling, and canopy leaching also plays an important role in the chemistry cycle of
forest ecosystems.

Although there have been studies on the spatio-temporal variability of rainfall

redistribution, most of these are limited to data of short-term monitoring over one-two
years or several months (Liu et al., 2019; Ziegler et al., 2009; Carlyle-Moses, 2004;





Marin et al., 2000). There are few studies exceeding several years experiments and
focusing on forest structural changes and rainwater interception (Grunicke et al., 2020;
Shinohara et al., 2015; Jackson, 2000). Long-term field monitoring studies are
considered to be valuable to gain insight into the temporal dynamics of forest
hydrological processes (Rodrigues et al., 2022; Sun et al., 2023; Levia and Frost, 2006).
Such studies can also contribute to identify patterns and trends in rainfall redistribution,
which is essential for predicting the long-term effect of water resource change on forest
ecosystems.

Therefore, in this study, we focus on the changing characteristics of throughfall

and stemflow in a subtropical forest succession sequence (pine forest→mixed forest→
monsoon evergreen broadleaf forest), based on long-term monitoring. Specifically, the
objectives are to analyze: (1) the changes of water volume of throughfall and stemflow
among the three forests, (2) the changes of water chemistry (TN, TP, $K^+$) of rainfall,
throughfall and stemflow among the three forests. We hypothesize that: (1) both
throughfall and stemflow in the broadleaf forest were characterized with high
variability compared to mixed forest followed by pine forest, (2) chemistry flux of
throughfall and stemflow changed over time with in order broadleaf forest > mixed
forest > pine forest. We aim to assess the variability of forest hydrological processes
from a long-term perspective to help predict future dynamic trends of water resources
in subtropical forest ecosystems.

**2. Materials and Methods**
*2.1. Study site*

This study was conducted at the Dinghushan Biosphere Reserve (23°09′ 21″ N ~

23°11′ 30″ N, 112°30′ 39″ E ~ 112°33′ 41″ E) located in Zhaoqing City, South China.
Dinghushan catchment consists of two streams both with 12 km length, which flow into
the West River (the main trunk of the Pearl River). According to the Köppen-Geiger
climate classification (Kottek et al., 2006), the study area belongs to tropical monsoon
climate (Cwa) with pronounced wet (April-September) and dry season (October-
March). The average annual temperature is 20.9 °C, and the annual rainfall and
evaporation are 1900 mm and 1115 mm, respectively. Dinghushan Biosphere Reserve
is covered with a complete horizontal succession series of three types of subtropical
forest, which is highly representative of the region (Zhou et al., 2011). Monsoon





evergreen broadleaf forest (BF) is 400 years old with typical tree species including
*Castanopsis chinensis (Spreng.) Hance*, *Schima superba Gardner & Champ*.,
*Cryptocarya concinna Hance*, etc. The mixed pine/broadleaf forest (MF) is a natural
succession with a coniferous broadleaf ratio of about 4:6, and 70–80 years old. The
main broadleaf tree species are *Schima superba Gardner & Champ*., *Castanopsis*
*chinensis (Spreng.) Hance*, and the coniferous species *Pinus massoniana Lamb*. The
pine forest (PF) planted before 1960 belongs to the primary succession community
where *Pinus massoniana Lamb* forms the only tree layer.

*2.2. Gross rainfall, throughfall and stemflow monitoring*
Atmospheric rainfall data was collected at Dinghushan Automatic Meteorological
Station from 2001–2022. The resolution of data recording was ±0.2 mm with a time
interval of 10 min. The raw data comprised annual rainfall amounts, as well as single
rainfall events with throughfall and stemflow measurements. Throughfall and stemflow
were collected in all three forest types and synchronously measured. Devices with
cross-shaped collectors (1.25 m$^2$) attached to reservoirs (1000 L) at the bottom were
used to collect throughfall. Three throughfall devices were randomly installed in each
forest field. Half-shell plastic tubes were installed around tree trunks attached to
reservoirs (1000 L) at the bottom to collect stemflow. A total of 24 trees with six tree
species were selected to measure stemflow volume. In detail, four tree species were
selected in the broadleaf forest, including *Acmena acuminatissima* (Blume) Merr. et
Perry (SF1), *Cryptocarya chinensis* (Hance) Hemsl. (SF2), *Gironniera subaequalis*
Planch. (SF3), *Schima superba* Gardn. et Champ. (SF4), with 3 repetitions respectively.
Three tree species were selected in the mixed forest, including *Castanea henryi* (Skam)
Rehd. et Wils. (SF5), *Schima superba* Gardn. et Champ. (SF6), *Pinus*
*massoniana* Lamb. (SF7), with 3 repetitions respectively. In the pine forest, *Pinus*
*massoniana* Lamb. (SF8) was selected as the monitoring subject with 3 repetitions.
Growth indicators of the selected trees have been recorded every five years since 2000:
tree height (m), DBH (cm), and crown area (CA, m$^2$). Forest structures have been
measured every five years since 2000: plant density (tree, shrub and herb), forest
canopy coverage (%) and LAI.



*2.3. Rainwater chemistry measurement*
For the measurement of rainwater chemistry, rainwater samples were from 2000,
2010 and 2022 in Dinghushan area. The samples of open rainfall, throughfall and
stemflow were manually collected for every one-month period, respectively. The
samples of open rainfall and throughfall were collected with three repetitions,
respectively and stemflow with four repetitions in the broadleaf forest, three repetitions
in the mixed forest and three repetitions in the pine forest. In total, 792 rainwater
samples (108 open rainfall, 324 throughfall and 360 stemflow) were collected.
Rainwater samples were defrosted and filtered through 0.45 μm polypropylene
membranes. Concentrations of total nitrogen (TN) and total phosphorus (TP) were
measured using ultraviolet spectrophotometer (Lambda 25, Perkin-Elmer), and ion
potassium ($K^+$) was measured using an inductively coupled plasma optical emission
spectrometer (Optima 2000, Perkin-Elmer), respectively. The origin data of TN, TP and
$K^+$ were processed into annual flux and monthly values by weighted average method,
$$C = \frac{\Sigma C_i * V_i}{\Sigma V_i} \tag{1}$$
where $C_i$ and $V_i$ are the concentrations of ions (mg $L^{-1}$) and water sample volume (L) in
each rainfall event, respectively.

*2.4. Statistical analysis and calculations*
The differences in throughfall and stemflow among different forests were assessed
using analysis of variance (ANOVA), followed by a Tukey test for multiple
comparisons between means. All statistical procedures were conducted with $\alpha = 0.05$
threshold for significance, in the IBM SPSS statistics 22.0 software (IBM Inc.).

**3. Results**
*3.1. Open rainfall*
Based on the 22 years rainfall dataset from the Dinghushan area, annual gross
rainfall ranged between 1370.0 and 2361.1 mm. In detail, approximately 80% of gross
rainfall appeared in the rainy season (April–September). Anomaly were revealed in the
temporal variability (coefficient of variation, *CV* of 16.6%) in annual rainfall (Fig. 1a).
In details, some remarkable negative values in 2003-2005, 2007, 2011 and 2021 and
positive values in 2006, 2008, 2015, 2018 and 2019 were found. Anomaly varied at -
426.4–476.8 mm and -258.0–471.4 mm in the rainy season and dry season, respectively.





By comparison, dry season experienced greater variation with *CV* of 40.4% than rainy
season with *CV* of 21.7%. Besides, annual raining days obviously tended to decrease
over time from 2012 to 2021(Fig. 1b). Based on five rainfall classifications, it was
shown among 22 years that rainfall <10 mm account for about 68.5% of total raining
days (2856), while rainfall >50 mm account for about 4.9%.

*3.2 Variability of throughfall*

Rainfall redistribution (throughfall + stemflow, TS) among the three forests all

experienced differing magnitude during 22 years (Fig. 2). Anomalies of TS revealed
that TS received below normal value similar with open rainfall. For throughfall ratio, it
varied significantly both at the event and interannual scales (Fig. 3a, b and c). The
median of annual throughfall ratio in the broadleaf forest varied between 60% and 120%
with *CV* of 13% from 2001–2022. The median of throughfall ratio in the mixed pine
and broadleaf forest varied between 80% and 110% with a *CV* of 10%. The median of
throughfall ratio in the pine forest varied between 59% and 110% with a *CV* of 11%.
Therefore, throughfall ratio was characterized by a relatively low variability over
annual-time scale (*CV* < 15%). Besides, some differences of throughfall ratio were
found among the three forest types based on rainfall classifications. For rainfall events
< 10 mm, throughfall ratio range in the broadleaf forest was 35%–70%, while in the
other two forest types it was 20%–85% (Fig. 3d). The mean value of throughfall ratio
was relatively small in the broadleaf forest (53.9%), though no significant difference
among the three forests (*P* > 0.05) were detected. For rainfall events <50 mm, no
significant difference of throughfall ratio among the three forest types was found (*P* >
0.05). However, the average values of throughfall ratio in the pine forest and the mixed
forest were both significantly larger than that in the broadleaf forest for rainfall
events >50 mm.

*CV* values of throughfall based on all the rainfall event classifications were drawn

in the Fig. 4a. Results showed that median *CV* of throughfall in the pine forest (15.2%)
was lower than that for the broadleaf forest (21.7%) and for the mixed forest (26.3%)
for rainfall events <10 mm. For rainfall events >10 mm, small differences of median
*CV* among the three forest types were shown. Meanwhile, *CV* values decreased with
the increasing rainfall events, eventually falling to 3.5%–4.3%. Besides, *CV* values of
throughfall based on interannual scale were drawn in the Fig. 5a, b and c. Annual *CV*
values among different forest types showed different fluctuations over time. The



medians of $CV$ in annual, rainy and dry seasons presented different order in different
years. According to linear fitting, significant negative correlations were found in the
median of $CV_{TF}$ in the mixed forest over time ($r = 0.63$, $P < 0.01$). In addition, fitting
result of in total 740 rainfall events in 22 years showed that $CV$ values of throughfall
significantly decreased with increasing gross rainfall (Fig. S1).

*3.3 Variability of stemflow*
Among 22 years, the median of annual stemflow ratio in the broadleaf forest varied
between 1.3% and 5.4% with a $CV$ of 56.2%. The stemflow ratio of mixed forest varied
between 1.5% and 4.4% with a $CV$ of 38.6%. In the pine forest, it varied between 0.3%
and 1% with a $CV$ of 50.9%. This indicated that the stemflow ratio was characterized
by an extremely high variability over time. Same to the seasonal throughfall ratio, the
medians of stemflow ratio in annual, rainy and dry seasons presented different orders
in different years. Besides, the stemflow ratio significantly changed among tree species
and among rainfall classifications (Fig. 3e). By comparison, stemflow ratios of the SF1
and SF2 trees in the broadleaf forest were both higher in all the tree species for the
rainfall events <50 mm. However, for strong events (>50 mm), the stemflow ratio of
the SF5 tree in the mixed forest was highest for all tree species, followed by the trees
in the broadleaf forest. For all the rainfall events, the stemflow ratio of SF7 in the mixed
forest and SF8 in the pine forest were both lower than that for other tree species.
$CV$ values of stemflow based on rainfall event classifications were drawn in the
Fig. 4b. By comparison, stemflow varied more than those of throughfall across rainfall
events, with $CV_{SF}$ values of 25%–130%. Median $CV$ of stemflow in the pine forest was
always lower (45%–68%) than that for the other two forest types (56%–120%). $CV$
values of stemflow based on interannual scale changed over time among different forest
types (Fig. 5d, e and f). The medians of $CV_{SF}$ in annual, rainy and dry seasons presented
different order in different years. By comparison, $CV_{SF}$ was always greater than $CV_{TF}$,
and interannual fluctuation of $CV_{SF}$ was also stronger than $CV_{TF}$. According to linear
fitting, significant negative correlations were found in the median of $CV_{SF}$ in the
broadleaf forest over time ($r = 0.73$, $P < 0.001$)). In addition, fitting result of in total
740 rainfall events in 22 years showed that $CV$ values of stemflow both significantly
decreased with increasing gross rainfall (Fig. S1).




*3.4. Rainwater chemistry*


Rainwater (open rainfall, throughfall and stemflow) chemical properties (TN, TP
and $K^+$ concentration) were measured in the three forest types, respectively. All of TN,
TP and $K^+$ values presented in order stemflow > throughfall > open rainfall (Fig. 6a, b
and c). However, changes of TN, TP and $K^+$ were different for the three forest types
among 2000, 2010 and 2022. For instance, in 2000 and 2010, TN values of throughfall
and stemflow decreased for both in order of pine forest > mixed forest > broadleaf forest,
while no such result could be confirmed in 2022. Similarly, TP values of throughfall in
broadleaf forest was 1.3 times higher than that in pine forest in 2022, while TP values
in pine forest was 6.8 time than that in broadleaf forest in 2000. $K^+$ values of stemflow
in 2010 (6.76 mg $L^{-1}$) and 2022 (6.22 mg $L^{-1}$) were higher for broadleaf forest than
those for pine forest (3.76 mg $L^{-1}$ and 2.46 mg $L^{-1}$), which was different from that in

2022.

TN, TP and $K^+$ fluxes of stemflow were < 10 kg $ha^{-1}$ $a^{-1}$, 0.2 kg $ha^{-1}$ $a^{-1}$, 6 kg $ha^{-1}$
$a^{-1}$, respectively, all lower than those of throughfall and open rainfall (Fig. 7d, e and f).
In the 2000, 2010 and 2022, TN flux (39.4–87.4 kg $ha^{-1}$ $a^{-1}$) was 1.2–1.8 times greater
than that of open rainfall, 3.3–28.0 times greater than that of stemflow. TP flux (1.1–
2.7 kg $ha^{-1}$ $a^{-1}$) was 1.0–2.3 times greater than that of open rainfall, 8.7–31.4 times
greater than that of stemflow. $K^+$ flux (21.5–59.2 kg $ha^{-1}$ $a^{-1}$) was 2.2–8.1 times greater
than that of open rainfall, 2.2–26.8 times greater than that of stemflow. In addition, TN,
TP and $K^+$ fluxes of stemflow increased with succession from primary to climax,
namely pine forest<mixed forest<broadleaf forest. Different from this, differences in
chemistry fluxes of throughfall was not found among different forests, neither among
different periods.
Besides, monthly chemistry concentrations in rainfall, throughfall and stemflow
showed distinct changes (Fig. 7). Monthly TN, TP and $K^+$ concentrations of rainfall
were always lower than those of stemflow for all trees. Monthly TN, TP and $K^+$ of
stemflow in the dry season were generally higher than in the rainy season. High monthly
TN concentrations of stemflow with SF6 of mixed forest and SF8 of pine forest were
found, especially in dry season with maximum TN concentrations of 27.59 mg $L^{-1}$ at
SF6 and 19.94 mg $L^{-1}$ at SF8, respectively. Differently, high monthly $K^+$ concentration
of stemflow at SF4 in broadleaf forest was found, with in dry season maximum $K^+$
concentration of 25.17 mg $L^{-1}$.




**4. Discussion**

*4.1. Open rainfall partitioned to throughfall and stemflow*

Studies in forests have confirmed that throughfall volume increased with increasing gross rainfall at event scale, accounting for 60%–80% of gross rainfall (Ji et al., 2023; André et al., 2011; Carlyle-Moses, 2004). This study, with 22 years of data, showed that annual rainfall changed over long-time scale and for different rainfall classifications (Fig. 1), which can directly affect annual throughfall. Throughfall ratio changed over time and showed different fluctuations among different forests (Fig. 3). During light rainfall events with rainfall amounts < 10 mm, a low proportion of raindrops would reach the ground as throughfall, as the tree canopy intercepts almost all the incoming raindrops. Specifically, high canopy coverage in broadleaf forest can reinforce raindrop intercept (Brasil et al., 2018; Ponette-González et al., 2010), consequently generating lower throughfall ratio than those in the mixed forest and pine forest (Fig. 3). During moderate rainfall events (10–50 mm), given that the intercept effect of the wetting tree canopy was weakened (Shinohara et al., 2015), throughfall ratio was in a high and steady state. As the gross rainfall increases further (> 50 mm), significant differences of throughfall ratio were found among the three forests. Throughfall ratio was significantly lower in the broadleaf forest than those in the other two forests. Likewise, such differences due to rainfall event class also appeared in other forest studies with stands such as beech, pine in monocultures and mixed pine-beech (Blume et al., 2022). Influenced by forest stand characteristics, throughfall therefore indicated different forest water budget.

Stemflow ratio of forests were variably controlled by tree species, on average accounting for about <10% of gross rainfall, even lower (<1%) (Sun et al., 2018; André et al., 2008; Crockford and Richardson, 1990). In our study site, the lowest stemflow (<1%) was collected in the pine forest, though weakly increasing with rainfall classifications (Fig. 3). Stemflow ratio in broadleaf forest was maintained at 5%–10% without the effect of rainfall amount seemingly. In detail, stemflow ratio of pine forest (SF8) was significantly lower than those of broadleaf forest (SF1~4). And in the mixed forest, broad-leaved trees (SF5 and SF 6) have larger stemflow than pine tree (SF7). However, for some rainfall events, extraordinary low proportion of stemflow in the broadleaf forest and extraordinary high proportion in the pine forest were caught. This





implied the role of rainfall conditions (e.g., intensity, duration) and tree species with
tree traits (e.g., branch angle), consistent with reported studies e.g., in evergreen forest
(Chen et al., 2019; Bruijnzeel et al., 2011) and pine forest (Crockford and Richardson,
1990). Moreover, ANOVA showed significant differences of stemflow ratio among tree
species and rainfall classifications ($P < 0.001$) (Table 1). This indicated that rainfall and
tree species simultaneously affect stemflow. Branch inclination angle, canopy cover,
tree height and DBH of tree species proved to be key factors in stemflow yield (Levia
et al., 2015).

Throughfall and stemflow were generally enriched in chemical concentration

compared with open rainfall due to leachable canopy/stem ion pools (Jiang et al., 2021;
Van Stan et al., 2017; Zimmermann et al., 2007). In our study, the concentration of $K^+$
in stemflow was 16 times higher than that in open rainfall and in throughfall reached
up to 11 times higher than open rainfall (Fig. 6). Similar results were also found in
artificial plantation (*Acacia mangium* and *Dimocarpus longan*) of South China (Shen
et al., 2013), in Oriental beech (*Fagus orientalis* Lipsky) trees in Northern Iran (Moslehi
et al., 2019), indicating strong $K^+$ leaching from canopy. Even so, throughfall was
generally characterized with high fluxes compared to open rainfall followed by
stemflow, it thus is the largest contributor to wet deposition. Meanwhile, TN flux of
throughfall was greatest in the pine forest in 2010, TP flux of throughfall was greatest
in the broadleaf forest in 2000, and $K^+$ flux of throughfall was greatest in the mixed
forest in 2010. It should be noted that the differences of rainwater chemistry shifted
over time among the three forests. Accordingly, throughfall and stemflow via canopy
and stem input soil is a significant contributor, and its long-term effect on ecosystems
needs more attention (Fan et al., 2021). After all, atmospheric wet deposition provides
nutrient requirement for ecosystems, but also imposes a considerable burden on the
ecosystems in general. For instance, N enrichment and P limitation have proven to have
different effect on soil carbon sequestration, microbial community composition and
forest productivity, especially in tropical and subtropical forest ecosystems with highly
weathered soils (Zheng et al., 2022; Li et al., 2016; Huang et al., 2012). Besides,
throughfall and stemflow was mainly characterized by low chemical concentrations in
the rainy season and high concentrations in the dry season. Primary reasons for seasonal
rainwater chemistry may be attributable to moisture source associated with frontal
weather systems and gradually depleting effect with increasing rainfall amount
(Dunkerley, 2014; Germer et al., 2007). The present study in subtropical forests and



previous studies in tropical forests and European temperate forests all exhibited variable
rainwater chemistry in throughfall and stemflow, both spatially and temporally
(Zimmermann et al., 2007; Staelens et al., 2006; Seiler and Matzner, 1995). In fact, the
chemical concentration of rainfall redistribution was also affected profoundly by
canopy and stem parameters of tree species (Tonello et al., 2021a; Chen et al., 2019).
In our study, some differences of TN, TP and $K^+$ were also found among SF1~SF8 due
to tree-species specific effect (Legout et al., 2016; De Schrijver et al., 2007).

*4.2. Long-term changes of rainfall in forests*

At the long-time scale of 22 years, the complexity of forest structure and rainfall

amount and their change exacerbated the spatio-temporal variability of throughfall and
stemflow. Firstly, interannual variability of forest structure (e.g., canopy coverage, leaf
area index) and tree parameters (e.g., height, DBH and CA) made throughfall and
stemflow distribution uncertain (Yue et al., 2021). From 2001 to 2022, changes in forest
structure were confirmed in all three forests, such as changes in plant density, canopy
coverage and LAI (Fig. 8). Throughfall ratio and stemflow ratio in the succession forest
systems all varied over time accordingly. Similarly, driven by forest structure (e.g., tree
density, species dominance), a six-year dataset from the Brazilian Atlantic Forest
showed that the spatial variability of throughfall over time was less stable (Rodrigues
et al., 2022). Besides, the variation of stemflow ($CV_{SF}$) was obviously larger than that
of throughfall ($CV_{TF}$) (Fig. 4), which probably was attributed to the differences of tree
species in stemflow (Fig. 3). For a forest succession, a 17 years' study showed that the
shift from monoculture Japanese red pine to mixture of red pine, evergreen oak and
theaceous tree made stemflow significantly increasing (Iida et al., 2005). Likewise, for
the forest succession in Dinghushan area, stemflow ratio in broadleaf forest and mixed
forest were both higher than that in tree-monospecific pine forest. High plant density
(tree and shrub) and LAI in broadleaf forest and mixed forest conduce to rainwater
interception of multi-canopy trees through more leaves and angled branches, which
potentially enhanced stemflow (Fig. 8). Indeed, some differences of rainfall
redistribution appeared in multi-layered vegetative structure. An experiment on
vegetation communities with a complex multi-layered structure found that interception
loss from shrubs was two-times higher than from trees, and smaller trees generated
stemflow more efficiently than the higher ones (Exler and Moore, 2022). Based on the
22 years' data from forest community survey in our study site, forest canopy parameters



(e.g., coverage and LAI) of trees and shrubs showed variation over time from 2001 to
2022 (Fig. 8). In the broadleaf forest, plant density of trees and canopy coverage of
shrubs showed a slight increment compared to the other two forests, though LAI was
decreasing. During this period, interannual throughfall ratio and stemflow ratio showed
significantly change over time (Fig. 3), implying the role of interannual variation of
forest structure in rainfall redistribution process.
Secondly, ongoing rainfall changes with different magnitude favor the different
levels of rainfall redistribution over time (Lian et al., 2022). At event scale, throughfall
and stemflow proportions of forests were both low with rainfall events <10 mm. The
variations of throughfall and stemflow were both larger for gross rainfall <10 mm than
events >10 mm. Rainfall threshold associated with the canopy interception capacity had
impact on throughfall and stemflow generation (Zabret et al., 2018; André et al., 2008;
Durocher, 1990). After the raindrop capacity of the canopy reached its peak, throughfall
and stemflow were documented to match the gross rainfall. Therefore, relatively low
proportions and high spatial variability appeared before rainfall threshold, and after that,
relatively high proportions and low variability until a stable level were observed in the
three forests. Moreover, at interannual scale, the raining days in different magnitudes
presented obvious fluctuation over 22 years (Fig. 1). This fluctuation of raining days
and its magnitude distribution potentially regulated the long-term changes of open
rainfall partitioned to interception loss, throughfall and stemflow. Consequently,
throughfall and stemflow, influenced by the comprehensive effect of rainfall regimes
and forest structures, presented spatiotemporal variability at different level (Fig. 3–6).
From a long-term perspective, changing in rainfall redistribution potentially makes
forest water and biogeochemistry budget more complex. Further knowledge of the
long-term accumulative effect of rainfall redistribution on forest water and chemistry
(e.g., soil and plant) is needed in the future.

**5. Conclusion**

The current study investigated long-term changing characteristic of rainfall
redistribution along a subtropical forest succession sequence with: pine forest (PF),
mixed pine and broadleaf forest (MF) and monsoon evergreen broadleaf forest (BF).
Firstly, in the valid 740 rainfall events throughfall ratio showed in order BF < MF < PF,
and stemflow ratio showed in order BF > MF > PF. The variation of stemflow was
higher ($CV$ >50%) than that of throughfall ($CV$ <25%). Secondly, 22 years' monitored



data showed that throughfall ratio widely changed between 30% and 90%, and
stemflow ratio changed between 0.1% and 10%. Annual gross rainfall considerably
changed over time, which directly induced the variable accumulation of annual
throughfall and annual stemflow.
For rainwater chemistry, stemflow was characterized with high TN, TP and $K^+$
concentrations compared to throughfall followed by open rainfall. Rainwater chemical
concentrations were lower in the rainy season than that in the dry season. However,
throughfall, characterized with high fluxes compared to open rainfall followed by
stemflow, is the largest contributor to wet deposition. Additionally, differences of the
rainwater chemical concentrations among the three forest types were confirmed over
time based on data from 2001, 2010 and 2022. On average, TN and TP fluxes of
throughfall presented in order BF < MF < PF, while $K^+$ flux of throughfall presented in
order BF > MF > PF.
The above results indicate that the water volume and chemistry in rainfall
redistribution process under forest are variable over time, and throughfall and stemflow
depend on the effect of rainfall and forest factors. This study thus provided insight into
the rainfall redistribution process by linking the long-term change of rainfall pattern
with a subtropical forest succession sequence.


*Code and data availability*. The data used to derive to the conclusions of the present
study are freely accessible. All the data were obtained from the CNERN dataset
(http://dhf.cern.ac.cn/meta/metaData).

*Author contributions*. WJZ: conceptualization, investigation, data analysis, writing,
visualization. TS and SS: reviewing, supervision. QMZ and CGW: resources, data
curation, WLH: reviewing, JXL: reviewing, funding acquisition, supervision

*Competing interests*. The authors declare that they have no conflict of interest.

*Disclaimer*. Publisher's note: Copernicus Publications remains neutral with regard to
jurisdictional claims made in the text, published maps, institutional affiliations, or any
other geographical representation in this paper. While Copernicus Publications makes
every effort to include appropriate place names, the final responsibility lies with the



authors.

*Acknowledgements*. Wanjun Zhang would like to acknowledge the financial support
from the CSC Fellowship.

*Financial support*. This research has been supported by The Key-Area Research and
Development Program of Guangdong Province (Grant No. 2022B1111230001), the
National Natural Science Foundation of China (Grant Nos. 42207158 and 32101342)
and the China Postdoctoral Science Foundation (Grant Nos. 2021M703259, 2021
M703260, 2021M693220).

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





**Tables**

**Table 1** Correlations between throughfall and stemflow and rainfall and forest factors

|  | Gross rainfall | DBH | CA | Height | LAI |
|---|---|---|---|---|---|
| Throughfall | 0.72*** | — | — | — | -0.58** |
| stemflow | 0.77*** | 0.65* | -0.75** | 0.54** | -0.55** |

DBH: diameter at breast height; CA: crown area; LAI: leaf area index. *$P < 0.05$, ** $P < 0.01$, *** $P <$
$0.001$






**Table 2** Analysis of variance (ANOVA) for throughfall and stemflow affected by rainfall
classifications and tree species

| Summary of ANOVA | Throughfall |  | Stemflow |
|---|---|---|---|
| Rainfall classification (R) | < 0.05 | Rainfall classification (R) | < 0.001 |
| Forest type (F) | < 0.001 | Tree species (T) | < 0.001 |
| R × F | 0.861 | R × T | < 0.001 |

$\alpha = 0.05$



**Figures**



**Fig. 1** (a) Anomaly of annual rainfall from 2001–2022 at the Dinghushan Biosphere Reserve in southern China, (b) annual raining days in five classifications



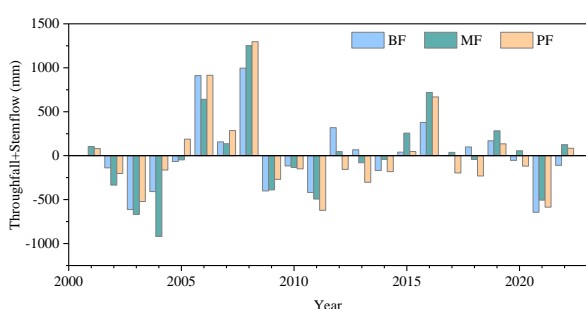

**Fig. 2** Anomaly of annual rainfall redistribution (throughfall + stemflow) in the broadleaf forest

(BF), mixed pine and broadleaf forest (MF) and pine forest (PF) from 2001–2022

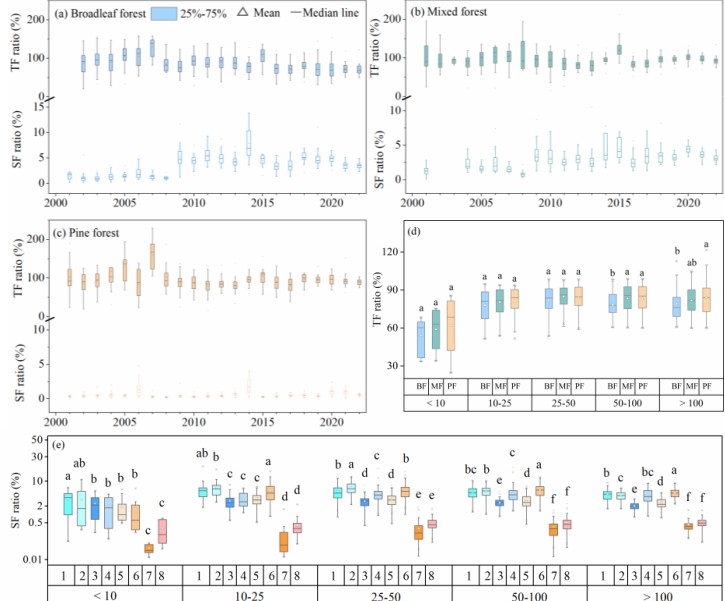

**Fig. 3** Box plots of throughfall ratio and stemflow ratio in (a) broadleaf forest, (b) mixed pine and
broadleaf forest and (c) pine forest from 2001–2022. Boxed plots of (d) TF ratio in the three
forests and (e) SF ratio for eight plant species based on the rainfall classifications (broadleaf
forest: *Acmena acuminatissima* (Blume) Merr. et Perry (SF1), *Cryptocarya chinensis* (Hance)
Hemsl. (SF2), *Gironniera subaequalis* Planch. (SF3), *Schima superba* Gardn. et Champ. (SF4);
mixed forest: *Castanea henryi* (Skam) Rehd. et Wils. (SF5), *Schima superba* Gardn. et Champ.
(SF6), *Pinus massoniana* Lamb. (SF7); pine forest: *Pinus massoniana* Lamb. (SF8). Different
letters indicate a significant difference at *P* < 0.05




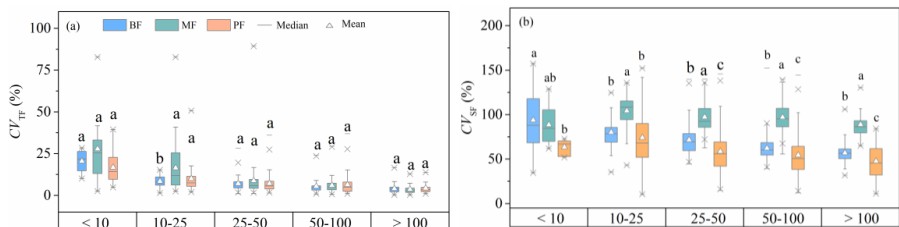


**Fig. 4** Box plots of coefficient of variation ($CV$, %) in (a) throughfall (TF) and (b) stemflow (SF) in
broadleaf forest (BF), mixed pine and broadleaf forest (MF) and pine forest (PF) based on the
rainfall classifications. Different letters indicate a significant difference at $P < 0.05$






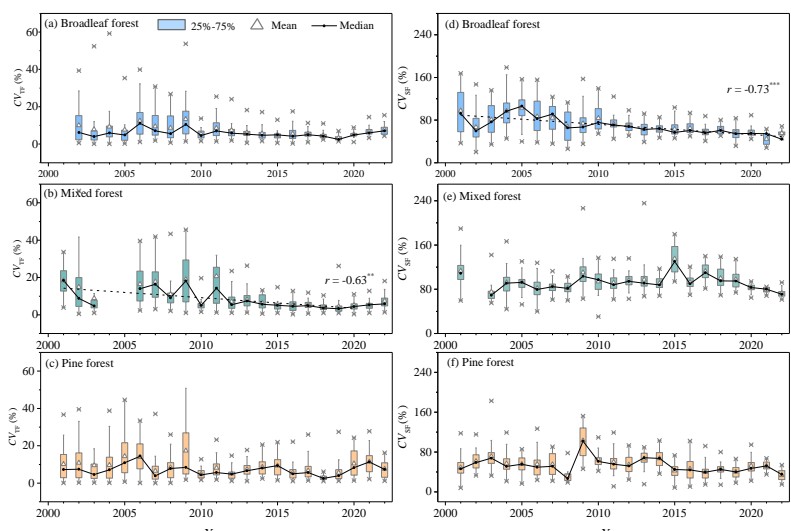


**Fig. 5** Box plots of coefficient of variation (*CV*, %) in (a, b, and c) throughfall (TF) and (d, e, and f)
stemflow (SF) in the three forests from 2001 to 2022. Medians of annual *CV* were fitted. *r*:
Pearson coefficient of correlation; *: $P < 0.05$, **: $P < 0.01$, ***: $P < 0.001$

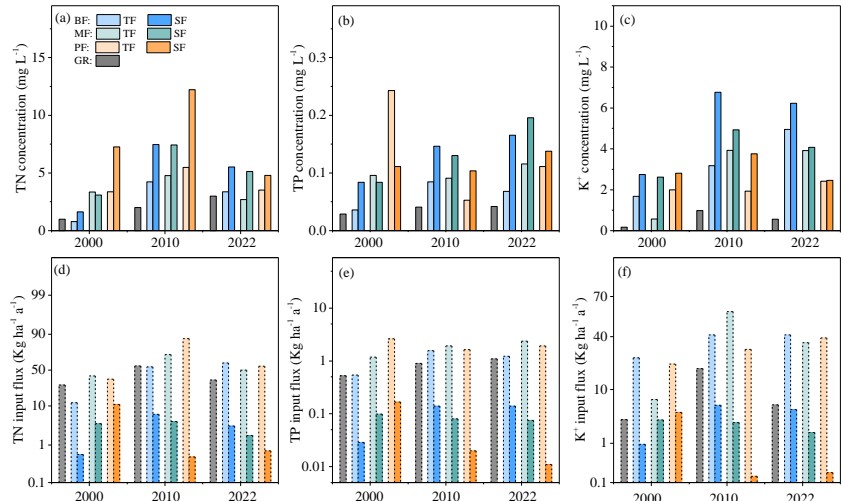


**Fig. 6** Concentrations and fluxes of TN, TP and K[+] of gross rainfall (GR), throughfall (TF) and

stemflow (SF) in the broadleaf forest (BF), mixed forest (MF) and pine forest (PF) in 2000,

2010 and 2022, respectively.



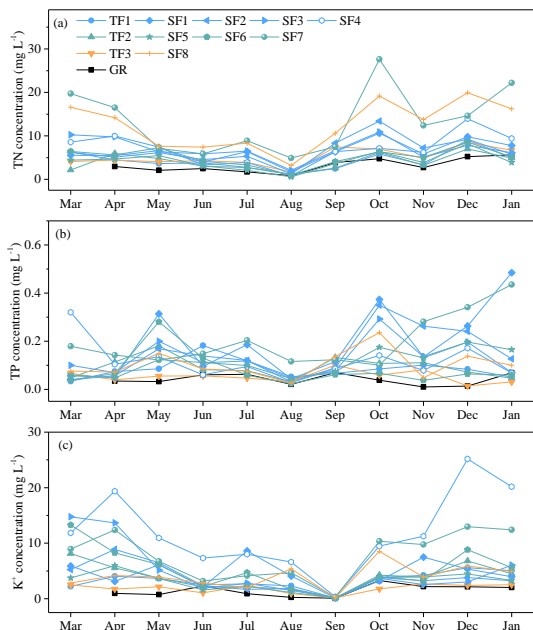


**Fig. 7** Monthly concentrations of (a) TN, (b) TP and (c) $K^+$ of throughfall (TF1) and stemflow (SF1,

SF2, SF3 and SF4) in the broadleaf forest, throughfall (TF2) and stemflow (SF5, SF6 and SF7)

in the mixed forest, throughfall (TF3) and stemflow (SF8) in the pine forest.



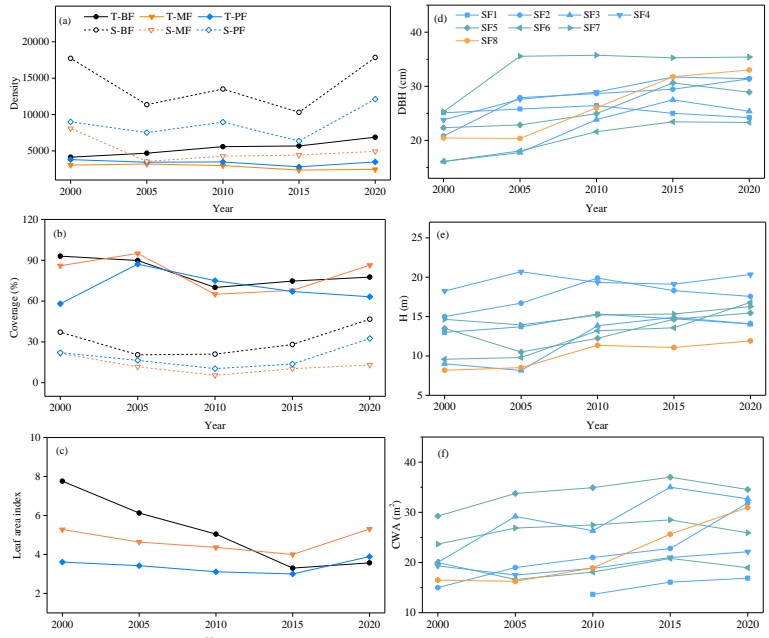

**Fig. 8** Plant density, canopy coverage and leaf area index of tree (T) and shrub (S) in the broadleaf
forest (BF), mixed forest (MF) and pine forest (PF), respectively. Diameter at breast height
(DBH), height (H) and crown area (CA) is given for eight stemflow-sampled trees, respectively.