# Peer review of "Rainfall redistribution in subtropical Chinese forests changes over"

_EGUsphere, 2023_

## Author Comment (AC1)

**Point-to-point responses to comments**

**RC1:**

Dear authors,

I am impressed by the wealth of data that the authors present in this study. An important issue to be considered in this investigation is the conducting of field research, which is not at all easy. The authors present the hydrological behavior and ion concentration in three forests and discuss the variability of this dynamics over 22 years.

Re: Dear referee,

Thank you very much, we are deeply appreciating your recognition of our study work. The constructive comments and suggestions absolutely can improve our manuscript. We carefully considered the comments and made corresponding revisions. Followings are point-to-point responses to your comments. We hope our revision can meet your expectation.

1. My remarks are more related to stemflow. I would like the authors to include basic information on DBH, Ht, and canopy area of the monitored trees. I also suggest including how trunk flow was measured.

Re: The information on DBH, etc. of 8 monitored tree species has been added in Table S2.

Measurement of stemflow: **The ratio of volume (mL) to canopy area ($cm^2$) is the stemflow (mm).**

**Table S2** Growth indicators of 8 monitored tree species in the forests

| Forest type | No. | | DBH | Height | Crown area |
|---|---|---|---|---|---|
| Broadleaf forest | SF1 | *Acmena acuminatissima* (Blume) Merr. et Perry | 23.6 ± 5.3 | 12.1 ± 2.1 | 18.1 ± 8.3 |
| | SF2 | *Cryptocarya chinensis* (Hance) Hemsl. | 28.8 ± 2.2 | 16.5 ± 1.4 | 28.5 ± 5.1 |
| | SF3 | *Gironniera subaequalis* Planch. | 23.8 ± 0.5 | 13.8 ± 0.7 | 26.3 ± 1.9 |
| | SF4 | *Schima superba* Gardn. et Champ. | 30.6 ± 1.7 | 20.4 ± 0.5 | 21.9 ± 2.9 |
| Mixed broadleaf-pine forest | SF5 | *Castanea henryi* (Skam) Rehd. et Wils. | 24.9 ± 3.2 | 12.2 ± 1.5 | 34.9 ± 1.0 |
| | SF6 | *Schima superba* Gardn. et Champ. | 21.6 ± 1.2 | 13.2 ± 0.8 | 18.1 ± 2.2 |
| | SF7 | *Pinus massoniana* Lamb. | 35.7 ± 1.5 | 15.2 ± 0.8 | 27.5 ± 7.4 |
| Pine forest | SF8 | *Pinus massoniana* Lamb. | 34.9 ± 1.2 | 10.2 ± 2.6 | 27.1 ± 7.2 |

DBH: diameter at breast height. 3 replications of each tree species, Mean ± SD

2. Did the authors believe that climate change might have caused any alterations in the data over time? Considering the ongoing global climate changes, it is conceivable that the authors may find it pertinent to include a statement regarding the potential influence of such environmental shifts on the observed data trends.

**Re:** Thank you for mentioning the topic of climate change and rainfall redistribution patterns, which prompts us to think and discuss it deeply. Currently, trends on the impact of climate change (global warming) on rainfall redistribution patterns are not well reported. We all know that throughfall and stemflow are part of rainfall and are an important player in the water cycle process. Based on the linkage of the water cycle to precipitation and temperature, we hypothesize that frequency of extreme events (heavy rainfall, droughts) and reduced biodiversity may affect rainfall redistribution and solute transport within forests, which in turn may affect the water cycle and biogeochemical cycles.

In this study, the 2008 rainfall data can be used as an example under extreme event. In 2008, extreme weather events occurred in China. In South China (subtropical region), freezing events of rain and snow occurred in the dry/winter season. In the wet/summer season, continuous heavy rain and typhoon events occurred, and the rainfall was larger than other years, with the annual rainfall of 2361.1 mm (22-year average annual rainfall of 1848.6 mm) (Fig. 1). At the same time, a total of 26 throughfall events were collected in 2008. According to the M-K test, the throughfall and stemflow trend of different forests presented different degree of disturbance under the background of mutation of open rainfall. In this process, the driving effect of forest structure and rainfall on throughfall and stemflow mutation is synchronous. Under the limited amount of data of extreme events, our study provides such hypothesis. More data and modeling are needed to support the relevant study about effect of climate change on rainfall redistribution in the future.

[Figure]

**Fig. 1** (a) and (b) annual rainfall and temperature of 2001-2022, respectively, (c) and (d) rainfall and temperature statistic of Mann-Kendall test, respectively. UF (Unadjusted Forward) > 0 indicate a continuous increasing trend ($P < 0.05$). The intersection points of UF and UB (Unadjusted Backward) is the mutation time point. Within the confidence interval [-1.96, 1.96], the variable presents a significantly mutation growth state ($P < 0.05$).

[Figure]

**Fig. 2** (a) and (b) Annual throughfall and stemflow in the broadleaf forest (BF), mixed pine and broadleaf forest (MF) and pine forest (PF) from 2001–2022, respectively, (c) ~ (h) rainfall and stemflow statistic of Mann-Kendall test, respectively. UF (Unadjusted Forward) > 0 indicate a continuous increasing trend ($P < 0.05$). The intersection points of UF and UB (Unadjusted

Backward) is the mutation time point. Within the confidence interval [-1.96, 1.96], the variable

presents a significantly mutation growth state at this time point ($P < 0.05$).

\*\*\*\*\*\*\*\*\*\*\*\*\*\*\*\*\*\*\*\*\*\*\*\*\*\*\*\*\*\*\*\*\*\*\*\*\*\*\*\*\*\*\*\*\*\*\*\*\*\*\*\*\*\*\*\*\*\*\*\*\*\*\*\*

---

## Author Comment (AC2)

**Point-to-point responses to comments**

**Referee #2:**

I believe this manuscript is of interest to the international readership, but have several revision requests as numbered below. Some are of major (possible rejection-level) concern (like the water balance often exceeds 100% of rainfall per Figures 2 and 3); however, others are of moderate-to-minor concern.

Intro is comprehensive and well-structured overview of the significance of rainfall redistribution in forest ecosystems, adeptly setting the stage for study's focus on long-term monitoring within a subtropical forest succession sequence. References are up-to-date and effectively highlight the relevance and timeliness of the research. The detailed introduction of interception loss, throughfall, and stemflow is suited to the scope of the study. And, identification of a research gap regarding the scarcity of long-term studies on forest structural changes and rainwater interception is compelling and justifies the necessity of the work.

Re: Dear referee,

We are deeply appreciating your recognition of our study work. The constructive comments and suggestions absolutely can improve our manuscript. We carefully considered the comments and made corresponding revisions. Followings are point-to-point responses to your comments. We hope our revision can meet your expectation.

(1) L157-160: Please provide a deeper rationale for your hypotheses, especially regarding the expected variability and chemical changes in throughfall and stemflow among different forest types (e.g., why broadleaf forest is expected to show higher variability?), could offer readers more insight into your theoretical framework.

Re: Thank you for insightful advice. We made such scientific hypotheses based on forest structure. In addition to rainfall factors, forest structure such as leaf area index, canopy coverage and plant density can affect the distribution of throughfall and stemflow. The vegetation community structure of the broadleaf forest is complex, and the canopy layer presents a multi-layer structure in the vertical direction. The pine forest has a single tree layer structure. By comparison, under the complex canopy structure, the splashing of canopy on raindrops during falling process is more frequent. Thus, we hypothesized that the throughfall and stemflow of broadleaf forest and have higher

variability than mixed forest and pine forest. Similarly, due to the complex canopy structure, rain leaching is also frequent, so we hypothesized that the leaching flux changes of broadleaf forest is greater than that of mixed forest and pine forest.

Study site section is a bit brief. Two suggestions:

(2) Readers may benefit from the inclusion of greater forest structure/diversity details. While you mention the ages and types of forests, additional information on forest structure (e.g., canopy height, basal area, stand density) and biodiversity (e.g., species richness, endemic species, understory composition) would better contextualize the systems under study.

Re: Thank you. Additional information on forest structure and monitored trees have been added in Table S1 and Table S2, respectively. As following,

**Table S1** Forest community structure and biodiversity

| Forest type | Community composition | $R_0$ | $H'$ | $J$ | $D$ |
|---|---|---|---|---|---|
| Broadleaf forest | Tree (*Syzygium acuminatissima*, *Cryptocarya chinensis*, *Gironniera subaequalis* Planch., *Schima superba*, *Aporosa yunnanensis*, etc.) | 13.66 | 2.69 | 1.04 | 0.73 |
| | Shrub (*Rhodomyrtus tomentosa* (Ait.) Hassk., *Baeckea frutescens* L., *Melastoma candidum* D. Don, etc.) | 18.03 | 2.66 | 0.93 | 0.75 |
| | Herb (*Eriachne pallescens* R. Br., *Pteris vittata* L., Ischaemum ciliare Retz., etc.) | 7.85 | 2.60 | 1.33 | 0.79 |
| Mixed broadleaf-pine forest | Tree (*Cryptocarya chinensis*, *Schima superba, Pinus massoniana*., etc) | 7.14 | 2.24 | 1.17 | 0.71 |
| | Shrub (*Litsea rotundifolia* var. *oblongifolia* (Nees) C. K. Allen, *Psychotria asiatica* L., *Ardisia quinquegona* Blume, etc.) | 8.80 | 2.26 | 1.05 | 0.70 |
| | Herb (*Gahnia tristis* Nees in Hooker & Arnott) | 4.80 | 1.69 | 1.19 | 0.60 |
| Pine forest | Tree (*Pinus massoniana.*) | 8.65 | 2.62 | 1.24 | 0.79 |
| | Shrub (*Blastus cochinchinensis* Lour., *Psychotria asiatica* L., etc.) | 14.83 | 3.08 | 1.16 | 0.83 |
| | Herb (*Tectaria harlandii* (Hook.) C. M. Kuo, *Alpinia oblongifolia* Hayata, etc.) | 8.20 | 2.29 | 1.16 | 0.71 |

$R_0$: Patrick richness, $H'$: Shannon-Weiner index, $J$: Pielou index, $D$: Simpson index

**Table S2** Growth indicators of 8 monitored tree species in the forests

| Forest type | No. | | DBH | Height | Crown area |
|---|---|---|---|---|---|
| Broadleaf forest | SF1 | *Acmena acuminatissima* (Blume) Merr. et Perry | 23.6 ± 5.3 | 12.1 ± 2.1 | 18.1 ± 8.3 |
| | SF2 | *Cryptocarya chinensis* (Hance) Hemsl. | 28.8 ± 2.2 | 16.5 ± 1.4 | 28.5 ± 5.1 |
| | SF3 | *Gironniera subaequalis* Planch. | 23.8 ± 0.5 | 13.8 ± 0.7 | 26.3 ± 1.9 |
| | SF4 | *Schima superba* Gardn. et Champ. | 30.6 ± 1.7 | 20.4 ± 0.5 | 21.9 ± 2.9 |
| Mixed broadleaf-pine forest | SF5 | *Castanea henryi* (Skam) Rehd. et Wils. | 24.9 ± 3.2 | 12.2 ± 1.5 | 34.9 ± 1.0 |
| | SF6 | *Schima superba* Gardn. et Champ. | 21.6 ± 1.2 | 13.2 ± 0.8 | 18.1 ± 2.2 |
| | SF7 | *Pinus massoniana* Lamb. | 35.7 ± 1.5 | 15.2 ± 0.8 | 27.5 ± 7.4 |
| Pine forest | SF8 | *Pinus massoniana* Lamb. | 34.9 ± 1.2 | 10.2 ± 2.6 | 27.1 ± 7.2 |

DBH: diameter at breast height. 3 replications of each tree species, Mean ± SD

(3) Given the long-term monitoring value of this study, please provide information on climate variability and trends. While the average climate conditions are provided, information on climate variability and any observed long-term climate trends (e.g., changes in rainfall patterns, temperature increases) would be pertinent.

Re: Thank you. According to your insightful advice, the information on rainfall and temperature were added, including seasonal and annual changes and Mann-Kendall test (Fig. 1).

**Based on the 22 years rainfall dataset from the Dinghushan area, annual gross rainfall ranged between 1370.0 and 2361.1 mm (Fig. 1). 78.0% of gross rainfall appeared in the wet season (April–September). Result of M-K test showed that rainfall was in a decreasing trend from 2001–2007 (UF < 0), and shifted into an increasing trend from 2012–2022 (UF > 0). Moreover, 2007–2009 and 2011 (the intersection of UF and UB) were the mutation time of rainfall trend ($P < 0.05$). Anomaly were revealed in the temporal variability (coefficient of variation, *CV* of 16.6%) in annual rainfall (Fig. 1a). Anomaly varied at -426.4–476.8 mm and -258.0–471.4 mm in the rainy season and dry season, respectively. By comparison, dry season experienced greater variation with *CV* of 40.4% than rainy season with *CV* of 21.7%. Besides, annual raining days obviously tended to decrease over time**

from 2012 to 2021(Fig. 1b). Based on five rainfall classifications, it was shown among 22 years that rainfall <10 mm account for about 68.5% of total raining days (2856), while rainfall >50 mm account for about 4.9%.

[Figure]

**Fig. 1** (a) and (b) rainfall and temperature in Dinghushan Biosphere Reserve in Southern China from 2001–2022, (c) and (d) annual rainfall and temperature, (e) and (f) rainfall and

temperature statistic of Mann-Kendall test, respectively. (g) Anomaly of annual rainfall from 2001–2022, (h) annual raining days in five classifications. UF (Unadjusted Forward) > 0 indicate a continuous increasing trend ($P < 0.05$). The intersection points of UF and UB (Unadjusted Backward) is the mutation time point. Within the confidence interval [-1.96, 1.96], the variable presents a significantly mutation growth state at this time point ($P < 0.05$).

Methods require greater details:

(4) Please detail the equipment for the Dinghushan Automatic Meteorological Station

Re: Automatic meteorological systems were used to measure atmospheric pressure (DPA501 gas-pressure meter), temperature (HMP45D sensor), relative humidity (HMP45D sensor), rainfall (SM1-1 pluviometer), etc. Datalogger (America Campbell, CR1000X) was used to control measurement sensors and store meteorological data.

(5) While the equipment used for collecting throughfall and stemflow is adequately described, additional information on the rationale behind the number of collectors and their placement within each forest type could provide insights into the sampling strategy's robustness. Why only 3 throughfall collectors per site (per L192)? This is very low replication given that it has long been known that throughfall has a high spatial variability.

Re: Thanks a lot for your insightful comments. We provided additional information, including collecting equipment of throughfall and stemflow (Fig. S1). As you can see, 3 replications (catch area 1.25 m$^2$ of each collector) for throughfall were set for each forest type, which is indeed a low replication, and may not be representative of the level of throughfall at the forest scale because a large number of studies have already demonstrated high spatial variability of throughfall. We acknowledge this as a fact and pity.

However, we would like to let the researchers know that the observations for throughfall and stemflow have been carried out since 1999, data on the longer time scales of the three forest types were well accumulated, which is a valuable data resource. When initiating this study over a long-time scale, we were already aware of the inadequacy (low replication) of this data set of throughfall. We thought about this carefully scientifically and logically and decided to maximize the use of these imperfect data (2001-2022), namely pay more attention to the temporal changes of throughfall over a long time series.

[Figure]

**Fig. S1** Collecting equipment and their locations of throughfall and stemflow

Besides, we referred to previous studies on the rainfall redistribution patterns of different forests. First, as to throughfall collector, throughfall was collected by different numbers of gauges with different surface areas, such as 3 gauges with 980 $cm^2$/per (Germer et al., 2006), 36 gauges with 95 $cm^2$/per (Loescher et al., 2002), 16 gauges with 254 $cm^2$/per (Fathizadeh et al., 2014), 24 gauges with 325 $cm^2$/per (Keim et al., 2018), 31 gauges with 378.5 $cm^2$/per (Rodrigues et al., 2022) and 5 gauges with 1.32 $m^2$/per (Blume et al., 2022). In this study, 3 gauges with 1.25 $m^2$/per for throughfall were set for each forest type. Second, as to spatial variability usually expressed in terms of coefficient of variation (CV), throughfall events covered five rainfall classifications (< 10 mm, 10-25 mm, 25-50 mm, 50-100mm, >100 mm), and their CV values decreased with the increasing rainfall (Fig. S2), which was consistent with the results of previous studies (Fathizadeh et al., 2014; Carlyle-Moses et al., 2004). Based on these previous studies and by comparison, we ensure that the Dinghushan throughfall dataset is within a reasonable range and the data are credible.

[Figure]

**Fig. S2** Coefficient of variation changing of throughfall ratio (a~c) and stemflow ratio (d~f) with rainfall volume. *r*: Pearson coefficient of correlation; *: $P < 0.05$, **: $P < 0.01$, ***: $P < 0.001$

**References:**

Blume, T., Schneider, L., & Güntner, A. (2022). Comparative analysis of throughfall observations in six different forest stands: Influence of seasons, rainfall-and stand characteristics. Hydrological Processes, 36(3), e14461.

Carlyle-Moses, D.E., Laureano, J.F., & Price, A.G. (2004). Throughfall and throughfall spatial variability in Madrean oak forest communities of northeastern Mexico. Journal of Hydrology, 297(1-4), 124-135.

Fathizadeh, O., Attarod, P., Keim, R. F., Stein, A., Amiri, G. Z., & Darvishsefat, A. A. (2014). Spatial heterogeneity and temporal stability of throughfall under individual *Quercus brantii* trees. Hydrological Processes, 28(3), 1124-1136.

Germer, S., Elsenbeer, H., & Moraes, J. M. (2006). Throughfall and temporal trends of rainfall redistribution in an open tropical rainforest, south-western Amazonia (Rondônia, Brazil). Hydrology and Earth System Sciences, 10(3), 383-393.

Keim, R. F., & Link, T. E. (2018). Linked spatial variability of throughfall amount and intensity during rainfall in a coniferous forest. Agricultural and forest meteorology, 248, 15-21.

Loescher, H.W., Powers, J.S., & Oberbauer, S.F. (2002). Spatial variation of throughfall volume in an old-growth tropical wet forest, Costa Rica. Journal of Tropical Ecology, 18(3), 397-407.

Rodrigues, A.F., Terra, M.C., Mantovani, V.A., Cordeiro, N.G., Ribeiro, J.P., Guo, L., ... & Mello, C.R. (2022). Throughfall spatial variability in a neotropical forest: Have we correctly accounted for time stability?. Journal of Hydrology, 608, 127632.

(6) The criteria for selecting the 24 stemflow trees and their representation of the broader forest sites would be valuable. Can the authors please provide information on

how these trees were deemed representative of their respective forest types?

Re: Thank you for advice. Here, we chose the eight tree species (24 trees) for two reasons: dominant species of tree layer and integrity of long-term data. First, community surveys show that the dominant tree species in the broadleaf forest in Dinghushan are *Syzygium acuminatissima*, *Cryptocarya chinensis*, *Gironniera subaequalis* Planch., *Schima superba*, *Aporosa yunnanensis*, etc. The dominant tree species in the mixed broadleaf-pine forest are *Cryptocarya chinensis*, *Schima superba* and *Pinus massoniana*. The dominant tree species in the pine forest is *Pinus massoniana.* Second, during the monitoring period (1999-), several trees were damaged due to extreme weather or abnormal growth, which prevented the stemflow from being properly collected. Thus, trees with missing data were not used when screening long-time series data.

(7) The methods used to measure the tree traits (L204-206) are absent. Please provide these details.

Re: Many Thanks. According your advice, the measurement methods were added:

Tree height was measured using laser range finder. Tape measure was used to measure the diameter of trees at a height of 1.3 m, namely DBH (diameter at breast height). CA (crown area): the laser rangefinder was used to measure the maximum diameter at the edge of the canopy, with multiple measurements at different points to ensure accuracy. Plant density: 25 plots of 20 m × 20 m (A1-A25 plots) were built on a plot of 1hm$^2$ to survey tree density. Then, 25 plots of 5 m × 5 m (B1-B25 plots) were randomly set on the A1-A25 plots to survey shrub density. Finally, 25 plots of 1 m × 1 m (C1-C25 plots) were randomly set on the B1-B25 plots to survey herb density. Canopy coverage: 25 observation plots (1 m × 1 m) were selected in the 1 hm$^2$ area of each forest type. The percentage of the surface area covered by plants to the total plot area is termed canopy coverage (%). LAI (Leaf area index) was measured using a LAI-2200 plant canopy analyzer with 90° view caps (Li-Cor Inc., USA). 10 observation points (distance about 10 m) were selected in the 1 hm$^2$ area of each forest type with 5 replications.

Results have a few oddities:

(8) Building on point 5 (throughfall under-sampling) – In comparing Figures 1a and 2, the net rainfall anomalies (throughfall plus stemflow shown in Fig 2) are much larger than shown for gross rainfall (in Fig. 1a). How can net rainfall anomalies be greater

than rainfall? This is also the case in Figure 3a-c, where throughfall fraction was often ~100% or more than gross rainfall. This suggests an overrepresentation of drip points (which often exceed 100% of rainfall), which would erroneously elevate the fraction of throughfall relative to rainfall. This is a rejectable error, in my opinion, unless there is another reason for the water balance exceeding 100% this often.

Re: Thanks a lot. We checked and reanalyzed the original data and found errors in the 2008 data. In 2008, extreme weather events occurred in China. In South China, freezing events of rain and snow occurred in the dry season. In the wet season, continuous heavy rain and typhoon events occurred, and the rainfall was larger than other years, with the annual rainfall of 2361.1 mm (Table S3). At the same time, a total of 26 throughfall events were recorded in 2008, of which four events had incomplete data (only one or two repetitions) due to the typhoon caused damage to the collectors. According to the measured data, throughfall and stemflow from 2001 to 2022 was overestimated and/or underestimated, consequently the wrong information was shown. In order to eliminate the bias, we decided to exclude the four events and used the valid data of 22 events with 198 (22*3 repetitions *3 forest types) to plot the throughfall and stemflow for 2001-2022 (Fig. 2).

**Table S3** Anomaly of annual rainfall, throughfall and stemflow in the three forest types from 2001-2022

| Year | Rainfall | Broadleaf forest | | Mixed forest | | Pine forest | |
|------|----------|------|------|------|------|------|------|
| | | TF | SF | TF | SF | TF | SF |
| 2001 | 1889.2 | | 30.4 | 1772.9 | 25.7 | 1724.9 | 11.1 |
| 2002 | 1818.9 | 1550.0 | 17.2 | 1401.5 | | 1632.5 | 12.8 |
| 2003 | 1344.7 | 1150.5 | 13.2 | 1089.1 | 13.1 | 1182.4 | 12.1 |
| 2004 | 1328.1 | 1242.1 | 18.4 | 954.2 | 19.4 | 1254.6 | 12.4 |
| 2005 | 1615.0 | 1513.1 | 22.3 | 1458.1 | 21.8 | 1517.3 | 13.9 |
| 2006 | 2227.6 | 1581.2 | 30.9 | 2000.5 | 32.2 | 2048.7 | 24.8 |
| 2007 | 1423.1 | 1377.8 | 17.1 | 1306.3 | 18.8 | 1379.8 | 9.0 |
| 2008 | 2361.1 | 2067.6 | 22.0 | 2186.6 | 20.8 | 2192.6 | 13.3 |
| 2009 | 1760.4 | 1264.7 | 90.8 | 1309.0 | 48.3 | 1564.5 | 15.0 |
| 2010 | 1735.8 | 1509.3 | 86.8 | 1531.9 | 55.3 | 1519.0 | 9.2 |
| 2011 | 1370.0 | 1249.6 | 91.5 | 1223.9 | 38.8 | 1214.7 | 10.5 |
| 2012 | 2028.3 | 1856.9 | 114.0 | 1684.0 | 67.0 | 1676.4 | 19.9 |

| 2013 | 2036.6 | 1652.9 | 101.9 | 1579.1 | 56.9 | 1522.3 | 33.7 |
|------|--------|--------|-------|--------|------|--------|------|
| 2014 | 1979.1 | 1457.0 | 97.3 | 1624.9 | 44.0 | 1645.4 | 30.1 |
| 2015 | 2182.0 | 1641.1 | 88.5 | 1870.4 | 71.1 | 1883.1 | 11.1 |
| 2016 | 1920.8 | 1693.2 | 100.1 | 1776.1 | 74.0 | 1749.6 | 22.6 |
| 2017 | 1860.8 | 1609.5 | 86.7 | 1673.1 | 70.3 | 1635.4 | 19.5 |
| 2018 | 2209.3 | 1664.8 | 119.7 | 1600.6 | 71.0 | 1599.7 | 22.7 |
| 2019 | 2158.7 | 1725.7 | 119.1 | 1895.7 | 67.9 | 1967.9 | 16.8 |
| 2020 | 1934.9 | 1543.5 | 109.4 | 1680.7 | 80.4 | 1534.8 | 35.4 |
| 2021 | 1506.8 | 1100.7 | 40.2 | 1203.1 | 49.4 | 1242.0 | 27.7 |
| 2022 | 1978.3 | 1517.7 | 84.1 | 1754.2 | 67.3 | 1722.0 | 26.2 |
| Mean | 1848.6 | 1522.3 | 68.3 | 1571.6 | 48.3 | 1609.5 | 18.6 |

[Figure]

**Fig. 2** (a) and (b) Annual throughfall and stemflow in the broadleaf forest (BF), mixed pine and broadleaf forest (MF) and pine forest (PF) from 2001–2022, respectively, (c) ~ (h) rainfall and stemflow statistic of Mann-Kendall test, respectively. UF (Unadjusted Forward) > 0 indicate a continuous increasing trend ($P < 0.05$). The intersection points of UF and UB (Unadjusted Backward) is the mutation time point. Within the confidence interval [-1.96, 1.96], the variable presents a significantly mutation growth state at this time point ($P < 0.05$).

(9) Why is the summary of throughfall fraction in Figure 3d so much lower than those in the previous panels? Panels a-c have y-axis values as high as 200% of rainfall, while

panel d has a y-axis max of 120%.

Re: Thank you. We checked raw data and recalculated throughfall ratio. We found that the different data sets were plotted in Figure 3a-c and 3d. The first dataset is the original data and wasn't averaged (Fig. 3a-c). The second data set was the average of three repeated TF ratios (Fig. 3d). In order to ensure consistent and valuable results, we finally chose the second data set and redraw the box plots.

[Figure]

**Fig. 3** Box plots of throughfall ratio and stemflow ratio in (a) broadleaf forest, (b) mixed pine and broadleaf forest and (c) pine forest from 2001–2022. Boxed plots of (d) TF ratio in the three forests and (e) SF ratio for eight plant species based on the rainfall classifications

(10) Finally, I'm unsure exactly what the 'take home message' is regarding temporal patterns in throughfall and stemflow. The authors conclude that "throughfall ratio widely changed between 30% and 90%, and stemflow ratio changed between 0.1% and 10%" (L476-478)…but what does this mean exactly? What is the theoretical or practical insight from this variability? And, given the undersampling of throughfall, how sure can we be that this variability is * real *?

Re: Many thanks. We thought deeply about your questions and recognized inaccurate

conclusion. So, according your comment, we revised the conclusion section. We hope these revisions meet your expectation.

**The current study investigated long-term changing characteristic of rainfall redistribution along a subtropical forest succession sequence with: pine forest (PF), mixed pine and broadleaf forest (MF) and monsoon evergreen broadleaf forest (BF). Firstly, in the valid 740 rainfall events throughfall ratio showed in order BF < MF < PF, and stemflow ratio showed in order BF > MF > PF. The variation of stemflow was higher (*CV* >50%) than that of throughfall (*CV* <25%). Secondly, 22 years' monitored data showed that throughfall ratio widely changed between 30% and 90%, and stemflow ratio changed between 0.1% and 10%. Driven rainfall and vegetation factors, interannual variability of throughfall and stemflow in the broadleaf forest was greater than those in the mixed forest and pine forest, which was different from that of annual open rainfall.**

For the under-sampling of throughfall, we were already aware of the inadequacy (low replication) of this data set of throughfall when initiating this study. We referred to previous studies on the rainfall redistribution patterns of different forests. First, as to throughfall collector, throughfall was collected by different numbers of gauges with different surface areas, such as 3 gauges with 980 $cm^2$/per (Germer et al., 2006), 36 gauges with 95 $cm^2$/per (Loescher et al., 2002), 16 gauges with 254 $cm^2$/per (Fathizadeh et al., 2014), 16 gauges with 254 $cm^2$/per (Fathizadeh et al., 2014), 24 gauges with 325 $cm^2$/per (Keim et al., 2018), 31 gauges with 378.5 $cm^2$/per (Rodrigues et al., 2022) and 5 gauges with 1.32 $m^2$/per (Blume et al., 2022). In this study, 3 gauges with 1.25 $m^2$/per for throughfall were set for each forest type. Second, as to spatial variability usually expressed in terms of coefficient of variation (CV), throughfall events covered five rainfall classifications (< 10 mm, 10-25 mm, 25-50 mm, 50-100mm, >100 mm), and their CV values decreased with the increasing rainfall (Fig. S2), which was consistent with the results of previous studies (Fathizadeh et al., 2014; Carlyle-Moses et al., 2004). Based on these previous studies and by comparison, we ensure that the Dinghushan throughfall dataset is within a reasonable range and the data are credible.

[Figure]

**Fig. S2** Coefficient of variation changing of throughfall ratio (a~c) and stemflow ratio (d~f) with rainfall volume. *r*: Pearson coefficient of correlation; *: $P < 0.05$, **: $P < 0.01$, ***: $P < 0.001$

**References:**

Blume, T., Schneider, L., & Güntner, A. (2022). Comparative analysis of throughfall observations in six different forest stands: Influence of seasons, rainfall-and stand characteristics. Hydrological Processes, 36(3), e14461.

Carlyle-Moses, D.E., Laureano, J.F., & Price, A.G. (2004). Throughfall and throughfall spatial variability in Madrean oak forest communities of northeastern Mexico. Journal of Hydrology, 297(1-4), 124-135.

Fathizadeh, O., Attarod, P., Keim, R. F., Stein, A., Amiri, G. Z., & Darvishsefat, A. A. (2014). Spatial heterogeneity and temporal stability of throughfall under individual *Quercus brantii* trees. Hydrological Processes, 28(3), 1124-1136.

Germer, S., Elsenbeer, H., & Moraes, J. M. (2006). Throughfall and temporal trends of rainfall redistribution in an open tropical rainforest, south-western Amazonia (Rondônia, Brazil). Hydrology and Earth System Sciences, 10(3), 383-393.

Keim, R. F., & Link, T. E. (2018). Linked spatial variability of throughfall amount and intensity during rainfall in a coniferous forest. Agricultural and forest meteorology, 248, 15-21.

Loescher, H.W., Powers, J.S., & Oberbauer, S.F. (2002). Spatial variation of throughfall volume in an old-growth tropical wet forest, Costa Rica. Journal of Tropical Ecology, 18(3), 397-407.

Rodrigues, A.F., Terra, M.C., Mantovani, V.A., Cordeiro, N.G., Ribeiro, J.P., Guo, L., ... & Mello, C.R. (2022). Throughfall spatial variability in a neotropical forest: Have we correctly accounted for time stability?. Journal of Hydrology, 608, 127632.

(11) The conclusions that are clearly stated appear to reinforce current knowledge without advancing theory. How might the following phrases be more clearly connected

to the authors' results so that they shed light on our current understanding?

Re: Thank you so much. We thought deeply about your questions and recognized inaccurate conclusion. So, according your comment, we revised them. We hope these revisions meet your expectation.

"Annual gross rainfall considerably changed over time, which directly induced the variable accumulation of annual throughfall and annual stemflow" – Rainfall becomes throughfall and stemflow… so, it is clear that variability in annual rainfall will directly affect variability in annual throughfall and stemflow

Re: We revised this sentence:

**Driven by rainfall and forest factors, interannual variability of both throughfall and stemflow in the broadleaf forest were greater than those in the mixed forest and pine forest, which was different from that of annual open rainfall.**

"stemflow was characterized with high TN, TP and K+ concentrations compared to throughfall followed by open rainfall" – This has been found in many forest types.

Re: We rewrote this sentence:

**Given the smaller proportion of open rainfall, stemflow chemical fluxes varied less among forest types and/or over time, though tree species exactly contribute to differences in stemflow chemistry. Nevertheless, its funnel effect on soil and plant over time still deserves more attention in the future.**

"throughfall, characterized with high fluxes compared to open rainfall followed by stemflow, is the largest contributor to wet deposition" – Given that this is an area and concentration game, it is well known that throughfall tends to dominate stand scale 'wet deposition' – but, what insights does this study provide across systems?

Re: Thank you so much. We thought deeply about this point and rewrote this sentence:

**For rainwater chemistry, differences of the clement flux in throughfall and stemflow among the three forest types were confirmed based on data from 2001, 2010 and 2022. On average, TN and TP fluxes of throughfall presented in order BF < MF < PF, while $K^+$ flux of throughfall presented in order BF > MF > PF.**
* * *